# A Normalized Fully Convolutionnal Approach to Head and Neck Cancer Outcome Prediction

## Abstract

Medical image classification performance worsens in multi-domain datasets, caused by radiological image differences across institutions, scanner manufacturer, model and operator. Deep learning is well-suited for learning image features with priors as constraints during the training process. In this work, we apply a ResNeXt classification network augmented with a UNet preprocessor sub-network to a public TCIA head and neck cancer dataset. The training goal is survival prediction of radiotherapy cases based on pre-treatment FDG PET-CT scans, acquired across 4 different hospitals. We show that the preprocessor sub-network in conjunction with aggregated residual connection improves over state-of-the-art results by 6% AUC (to 76%) while having less training parameters and not requiring segmentation annotations.

**Keywords:** Classification, head and neck cancer, deep learning, PET-CT, UNet-FCN, multi-domain, radiotherapy, outcome survival prediction

## 1. Introduction

Cancer treatment planning remains a long process for the patient: from pre-treatment staging to post-therapy follow-up, many factors could have changed that can impact the effectiveness of the treatment. One of the key decisions of the physician is the choice of line of therapy, for which automatic outcome prediction can be beneficial. In head and neck cancer cases, positron emission tomography with fluorodeoxyglucose integrated with computed tomography (FDG PET-CT) for diagnosis and treatment planning (Castaldi et al., 2013) can be used as inputs to deep learning-based medical image analysis models.

In previous works using this Cancer Imaging Archive (TCIA) dataset (Vallières et al., 2017), random forests were used to classify overall survival (OS) based on a combination of both PET and CT extracted radiomics features and clinical information (Vallières et al., 2017). More recently, an end-to-end convolutional neural network (CNN) was used to successfully predict radiotherapy outcomes using only the planning CT scans as input using the same dataset (Diamant et al., 2019).

In this work, we show that combining PET and CT image inputs improves binary classification performance. Furthermore, our model upgrades the simple CNN with aggregated residual connections following the ResNeXt approach (Xie et al., 2016) and a UNet preprocessor based on Drozdzal et al. (2018). We show that these three modifications in data input and network architecture improves our model's performance while reducing the number of training parameters required.

## 2. Methodology and Experiments

In this study, the training data consisted of 298 head and neck cancer patients acquired from 4 different institutions in Quebec. Each patient had a pre-radiotherapy FDG PET-CT scan. Both PET and CT volumes were converted to 2D images using largest primary GTV lesion area slice selection. Images were normalized to 0 mean and unit standard deviation. The dataset was split into 3:1:1 training, validation and testing sets.

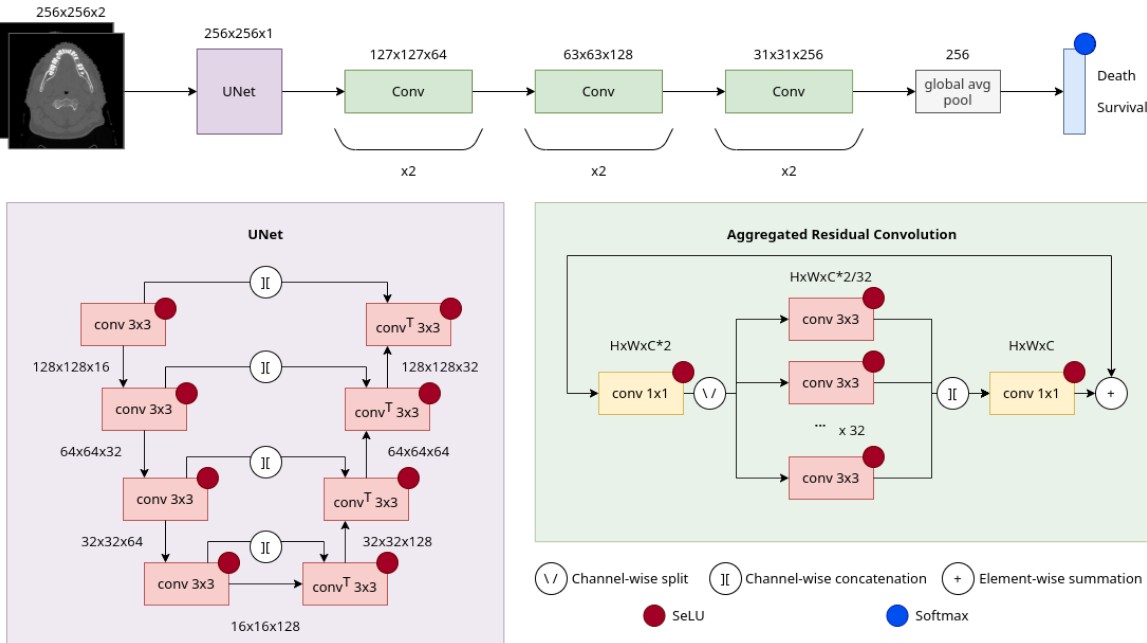

Figure 1: Proposed model architecture (top). The input consists of a 2 channel PET-CT image that is initially passed through a UNet (lower left). Downsampling uses convolutions with stride 2 while upsampling uses transposed convolutions. The output is then fed to an 18-layer deep CNN. Aggregated residual convolutional blocks (lower right) are repeated twice before being downsampled by setting the stride to 2. Classification is performed by taking the output vector with 256 features through a fully connected layer with softmax activation.

Our end-to-end binary classification model consists of two parts: a UNet sub-network (Ronneberger et al., 2015; Drozdzal et al., 2018) and a fully convolutional classifier shown in Figure 1. The UNet consists of 4 downsampling blocks followed by 4 upsampling blocks. Each block is composed of a 3x3 convolution layer with SeLU activation (Klambauer et al., 2017) and uses strided convolutions to change the output dimension. Output features of each downsampling block are concatenated with input features of each corresponding upsampling blocks.

The CNN classifier is inspired by the ResNeXt architecture (Xie et al., 2016), with 2 3x3 bottleneck layers with a filter growth factor of 2 and cardinality of 32 and residual

connections around each. Downsampling is done using strided convolutions in the first layer of each block. Global average pooling (He et al., 2015) and a fully connected layer are used to output the binary survival class.

Training is performed using categorical cross-entropy loss for classification (0: survival, 1: death) using the Adam optimizer (Kingma and Ba, 2014) with a learning rate of 0.0006 for 100 epochs and a batch size of 8. Training set inputs are oversampled to an even distribution of positive and negative samples to mitigate data imbalance. The code is implemented in Keras and trained on a GeForce RTX 2080 Ti for 1 hour.

## 3. Results and Conclusions

Binary classification results are monitored using the area under the receiver operating characteristic curve (AUC). Table 1 compares the results of our model with previous state of the art models on the same dataset, along with iterative proposed improvements.

Compared to the 5-layer CNN by Diamant et al. (2019), the 18-layer ResNeXt contains less than a third of the parameters, but reduces performance by 5% AUC. By introducing the the UNet preprocessor, this performance is recovered and matches the state of the art results of 70% AUC. Even with this addition, our proposed model contains less training parameters. As was shown by Drozdzal et al. (2018), the UNet acts as a image normalizer improving task-specific performance in a learned fashion. Whereas in their case it was designed for segmentation, our results of the UNet-ResNeXt for classification beating state of the art attests to its usefulness. Indeed by including a secondary PET input, the output can be considered can be considered an image embedding, fusing features from both scans on top of per modality normalization.

Table 1: Classification performance of the proposed models compared to state of the art results on the Head and Neck FDG PET-CT TCIA Dataset. The first model consists of a random forest for radiomics features selection followed by another random forest for classification. The second model consists of a CNN with 3 convolutions/PReLU/max-pooling layers and 2 fully connected layers. The last three models show ablation results of our proposed UNet 18-ResNeXt trained on PET and CT images.

| Model | Inputs | Parameters | AUC |
|---|---|---|---|
| Vallières et al. (2017) | Radiomics-CT + clinical data | - | 0.65 |
| Diamant et al. (2019) | GTV-masked CT | 930 146 | 0.70 |
| ResNeXt | CT | 291 874 | 0.65 |
| UNet-ResNeXt | CT | 683 410 | 0.70 |
| UNet-ResNeXt | PET + CT | 683 650 | 0.76 |

Thus, after training on both CT and PET images, our proposed model has overall less total parameters (683 650 < 930146) and improves in AUC by 6 percentage points over the state of the art. Finally, our model can be used without requiring manual GTV segmentation annotations, which remains time consuming for radiologist to generate.

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
