# OpenReview forum: "A Fully Convolutional Normalization Approach of Head and Neck Cancer Outcome Prediction"
_MIDL.io/2020/Conference — MIDL 2020_

### Official Review · AnonReviewer1 · 2020-02-25
**Nice extension of existing method, good validation and results**

**Rating:** 4
**Confidence:** 4

**Review:**

The paper evaluates the performance of a model based on a UNet pre-processor followed by a ResNeXt classifier for survival prediction in head and neck cancer patients. The proposed model uses a 2 channel input consisting of corresponding slices from CT and PET volumes from a publicly available dataset.

This is a well-written paper, and the descriptions of the method and experimental setup are clear and unambiguous. The results apparently outperform the state-of-the-art for this application and the model uses fewer parameters. Overall I am happy for this paper to be accepted but there are a few questions that need clarifying.

First, in Table 1, there is a reduction in performance of 5% AUC between the Diamant et al method and the basic ResNeXt model. The obvious question is whether this is due to the different architecture or the different input (i.e. GTV masked CT slice and full CT slice). Can the authors comment on this?

Also regarding Table 1, and depending on the answer to the first question, would it be possible to get even better results by combining the Diamant et al model with the UNet pre-processor and/or the PET data as input?

Other specific suggestions:
•	Title: “Convolutionnal” should be “Convolutional”
•	Section 3, paragraph 2: “the the” - remove repetition
•	Section 3, paragraph 2: “can be considered can be considered” – remove repetition

---

### Official Review · AnonReviewer4 · 2020-03-11
**Interesting study and need more ablation study**

**Rating:** 3
**Confidence:** 5

**Review:**

Summary:
The authors proposed a UNET preprocessor and a ResNeXt classification network for survival prediction. Improved performance is observed when using UNet-ResNeXt and PET+CT as input.  This is an interesting study and provides possible insights for other survival analysis during their model developments. But I think the authors need to do more ablation study and provide training details to let readers understand and use their work.

Major Concerns:
1. The authors claimed that the proposed model without requiring manual GTV segmentation annotations, but it seems the model needs to select largest primary GTV slice. If there are no GTV masks, how can you select largest GTV slice ?

2. It seems PET can help improve results a lot. Have you tried model only with PET ?

3. How did you compare with other baselines ? Are you using official split from Diamant et al. (2019) ?

4. Did you use any data augmentation during training ? Did you perform early stop because you have a validation set ?

5. What is the data distribution for death and survival ? How about Specifcity and Sensitivity ?

---

### Official Review · AnonReviewer3 · 2020-03-11
**U-Net preprocessing help improve accuracy of outcome prediction of hand and neck cancer**

**Rating:** 3
**Confidence:** 4

**Review:**

This paper performs hand and neck cancer outcome prediction by ResNeXt with a U-Net for preprocessing. However, the idea of Using a trainable U-Net as a preprocessing step is preciously used in segmentation task, so the methodological contribution is small.

In addition, the experiments are not very convincing. With the full proposed architecture, UNet-ResNeXt has the same AUC with a previous study (Diamant et al. 2019). A reduction of a small proportion of parameters is a very small advantage. Of course, with the addition of PET image, the AUC is increased to 0.76, but I think this is a natural result, considering the significant role of PET in staging the tumor. It’s a natural thought that the performance of (Diamant et al. 2019) can be equally improved by incorporating PET images.  Without the need of GTV mask may be an advantage of the proposed method over (Diamant et al. 2019), but this is uncertain without direct comparison of accuracy.

The paper is generally well written and easy to understand, but the first two sentences of Abstract are a little misleading. From the first sentence, it seems that this paper wants to deal with the problem originated from multi-domain datasets. However, there is no study about cross validation among different hospitals, though images from four hospitals are used. Regarding the second sentence, I don’t see any “priors as constraints” in training in the experiment.

---

### Official Review · AnonReviewer2 · 2020-03-13
**Preliminary results adequate for an abstract subject to fixes.**

**Rating:** 3
**Confidence:** 5

**Review:**

This paper proposes a CNN approach for head and neck cancer outcome prediction. A better motivation of the method is needed and clear statements of what is shown in the experiments.
More detailed comments are provided in the following:

I think that the main message of this paper is in the introduction “In this work, we show that combining PET and CT image inputs improves …” This is not in the abstract while it seems to be the only point made here.

In 1. Introduction: “We show that these three modifications …” is not supported by the experiments/results.

No evaluation is made on new centers ? unlike sort of motivated in the abstract.

What is the motivation for UNet + CNN? It is only mentioned in conclusion and should be given before.

It seems like the validation set is not used. Early stopping is not performed on the validation set?

There are typos in Section 3.

In the introduction, the radiomics method of Vallières et al. is mentioned with CT and PET but only with CT in Table 1.

---

### Meta-Review · Area_Chair1 · 2020-03-30
**MetaReview of Paper145 by AreaChair1**

**Rating:** 3

**Metareview:**

The paper evaluates the performance of a model based on a UNet pre-processor followed by a ResNeXt classifier for survival prediction in head and neck cancer patients, using CT and PET/CT images.

The paper is well-presented on the whole, the ideas are up-to-date and clearly described,  and the ablation study is interesting.

However, there is some slight caveat regarding the experimental results. As has been noted by all reviewers, the proposed approach is compared to state of the art methods with different inputs, and on a different dataset. So there is no conclusive evidence as to whether the proposed approach is superior to existing ones.

Some reviewers also note that the  very beginning of the abstract (2 first sentences) that may be misleading and should be rewritten, and that additional details regarding training are missing.

**Paper Type:**

validation/application paper

---

### Decision · Program_Chairs · 2020-04-11

Accept